# Imaging Review of the Lung Parenchymal Complications in Patients with IPF

**DOI:** 10.3390/medicina55100613

**Published:** 2019-09-20

**Authors:** Elisa Baratella, Ilaria Fiorese, Cristina Marrocchio, Francesco Salton, Maria Assunta Cova

**Affiliations:** 1Department of Radiology, Azienda Sanitaria Universitaria Integrata di Trieste (ASUITS), 34100 Trieste, Italy; 2Department of Pneumology, Azienda Sanitaria Universitaria Integrata di Trieste (ASUITS), 34100 Trieste, Italy

**Keywords:** IPF, consolidation, ground glass, infection, lung cancer, acute exacerbation, HRCT

## Abstract

Idiopathic pulmonary fibrosis (IPF) is a chronic, pulmonary-limited, interstitial lung disease with a poor prognosis. This condition is characterized by different clinical scenarios, ranging from the most typical slow and progressive deterioration of symptoms to a rapid and abrupt decline of lung function. Rapid worsening of clinical course is due to superimposed complications and comorbidities that can develop in IPF patients, with a higher incidence rate compared to the general population. These conditions may require a different management of the patient and a therapy adjustment, and thus it is fundamental to recognize them. High Resolution Computed Tomography (HRCT) is sensitive, but not specific, in detecting these complications, and can evaluate the presence of radiological variations when previous examinations are available; it recognizes ground glass opacities or consolidation that can be related to a large spectrum of comorbidities, such as infection, lung cancer, or acute exacerbation. To reach the final diagnosis, a multidisciplinary discussion is required, particularly when the clinical context is related to imaging findings.

## 1. Introduction

Idiopathic pulmonary fibrosis (IPF) is a chronic and progressive interstitial lung disease with a poor prognosis and a median survival rate of 2–3 years after diagnosis [1].

Several complications and comorbidities may occur in patients with IPF, sometimes with a fatal outcome. These conditions include infection, malignancy, pulmonary hypertension, pulmonary embolus, acute exacerbation, drug-induced pneumonitis, and cardiogenic pulmonary oedema [1,2,3].

High Resolution Computed Tomography (HRCT) is fundamental in diagnosing IPF, but also in detecting these complications and comorbidities that can superimpose on the underlying disease.

In this review, we will discuss the imaging findings of the most common parenchymal complications.

## 2. Imaging

According to the more recent guidelines of American Thoracic Society/European Respiratory Society/ Japanese Respiratory Society/ Latin American Thoracic Association (ATS/ERS/JRS/ALAT) and the Fleischner Society White Paper [4,5], high resolution computed tomography plays a central role in the diagnosis of diffuse interstitial lung diseases, particularly in patients with a clinical suspect of IPF. If a typical (Figure 1) or probable pattern of usual interstitial pneumonia (UIP) (Figure 2) is recognized on HRCT, in association with the appropriate clinical setting, the diagnosis of IPF can be made without surgical biopsy.

Patients with IPF have a higher risk of comorbidities and lung cancer compared with the general population [1].

HRCT has a role in establishing both acute and chronic parenchymal complications in patients with IPF and, in particular, in detecting areas of parenchymal consolidation or ground glass opacities, which can be an expression of infection, lung cancer, or acute exacerbation [6]. The differential diagnosis among these conditions could be more easily made by the radiologist if clinical data and previous Computed Tomography (CT) examination are available.

### 2.1. Infection

Patients with IPF may experience pulmonary infections with a higher frequency than patients without IPF. Lung infections are the most frequently occurring comorbidity and are a common cause of hospitalization [7]. It could be difficult to make a differential diagnosis with an acute exacerbation. Some studies suggest that chronic viral infections may have a role in the pathogenesis of IPF, in particular the Epstein–Barr virus and the latent human herpesvirus [8,9,10]. Moreover, the role of bacterial infections during infancy, especially from *Staphylococcus* and *Streptococcus* species, is being investigated [11,12,13].

The most frequently observed pathogens are *Mycobacterium* species and *Aspergillus* species, along with opportunistic infections such as *Pneumocystis jirovecii* [2]. The structural changes induced by IPF in the lungs and the impairment of the local host immunity create a favorable environment for mycobacterial colonization. Moreover, concomitant immunosuppressive therapies facilitate the reactivation of tuberculosis [14]. Aspergillosis results from the saprophytic colonization of pre-existing lung cavities.

Infections can be occasionally clearly diagnosed, as CT appearance is frequently non-specific, and pre-existing background of parenchymal alterations may mask or alter the typical radiographical pattern of these pathologies. An example can be given by the opportunistic infection of *P. jirovecii*, which typically manifests on HRCT as bilateral, symmetric ground glass opacity, consolidation, and septal thickening with a perihilar and upper lobe distribution and a subpleural lung sparing. Nevertheless, in IPF *P. jirovecii*, infection can be undistinguishable from an acute exacerbation of IPF or the CT pattern, showing no change compared to the previous examination, even in a clinical context of infection [2] (Figure 3).

Tuberculosis has been reported to occur more frequently in IPF patients than the general population [15,16]. A study by Shachor et al. found that the incidence of positive cultures for *Mycobacterium tuberculosis* in chronic interstitial lung diseases was 4.5 times that of the general population. In a study by Park et al. including 795 IPF patients, pulmonary infections with *Mycobacterium tuberculosis* and nontuberculous mycobacterium were found in 35 (4.4%) and 16 (2.0%) of patients, respectively. These rates were higher than those in the general population [14]. Patients with IPF are more prone to develop Tuberculosis (TB) reactivation with HRCT, with a peripheral mass-like lesion being shown to exist in more than half of patients, sometimes resembling lung cancer in IPF (Figure 4), but sometimes presenting as subpleural nodules [15]. In 33% of cases, a segmental or lobar coalescent consolidation with or without necrotizing cavitation may be observed, infrequently associated with the typical TB reactivation patterns (patchy multifocal consolidation and centrolobular nodules). It is mandatory to make a correct diagnosis to start an appropriate therapy.

Infections of *Aspergillus* species in IPF patients may result in the development of aspergilloma, typically developing in pre-existing lung cavities or in a chronic airway invasive aspergillosis. CT can effectively detect early changes, demonstrating the fungal fronds originating from the cavity wall in early stages, with the subsequent detachment and coalescence of the cavity itself resulting in the typical air crescent sign [2,17,18] (Figure 5).

In patients with IPF and bacterial infection, part-solid nodules (Figure 6) or ill-defined nodules (Figure 7) may be seen. The same radiological findings can be present in lung cancer; therefore, in some cases, a correct differential diagnosis is challenging. A Japanese cohort study found that, among patients with IPF that require hospitalization, Gram-negative species are the most common pathogens involved, in particular *Haemophilus influenzae* [19]. This is in contrast with community-acquired infections, in which Gram-positive bacteria are more commonly isolated.

### 2.2. Lung Cancer

Patients with IPF have a five-fold increase of developing lung cancer when compared to the general population [20,21], with an incidence rate of 11 cases per 100,000 persons per year. A large meta-analysis estimated a rate of lung cancer prevalence in IPF patients of 13,54% [22]. Some risk factors have been identified, such as age, male sex, and smoking history [22,23].

Both squamous cell carcinoma and adenocarcinoma have been observed, with a higher prevalence of the former [24,25,26]. Moreover, IPF patients experience synchronous tumors with a higher prevalence compared to non-IPF patients [27].

The pathogenesis remains unknown, and repeated episodes of cellular damage caused by inflammatory mediators have been hypothesized. This cellular damage may cause genetic injury to the respiratory epithelium, thus leading to cellular atypia, metaplasia, dysplasia, and finally carcinoma. Considering the already poor survival rate of IPF patients, it is difficult to determine the impact of the diagnosis of lung cancer on patients’ prognosis; moreover, papers reporting survival among IPF patients with lung cancer are limited by small sample sizes [24,28,29]. A study by Ozawa et al. reported a median survival time after lung cancer diagnosis in IPF of 13.1 months [20]. A retrospective analysis of 632 patients with IPF found a 1 year all-cause mortality rate after lung cancer diagnosis of 53.5%, confirming an overall poor survival [30].

On CT, lung cancer typically manifests as an ill-defined rounded lesion that can mimic an air-space consolidation (Figure 8), or presents as a nodular lesion (Figure 9), making it difficult to distinguish from other entities. In most instances, lung cancer develops in fibrosis areas, typically within or abutted areas of basal and peripheral honeycombing, with a typical prevalence for the lower lobes, compared to the general population where it generally manifests in the upper lobes [31,32].

Among invasive adenocarcinomas of the lung, invasive mucinous adenocarcinoma (previously known as mucinous bronchoalveolar carcinoma) may appear on CT as a ground glass opacity, which may have consolidative areas [33]. As shown in Figure 10, this pattern of CT presentation, in the context of a fibrotic lung parenchyma, may mimic an inflammatory area, making a differential diagnosis difficult. Therefore, a malignancy should always be considered as a possible diagnosis in case the lesion does not respond to corticosteroid and antibiotic therapy.

In a CT scan, it could be difficult to distinguish areas of dense fibrosis from lung carcinoma (Figure 11), thus a follow-up is required to evaluate the progression of this consolidation [34].

Analogous pattern on CT may be seen in pulmonary infarction and in TB reactivation. It has been largely demonstrated in the literature that patients with IPF are more prone to suffer from acute thromboembolism, thus developing pulmonary infarction [35,36]. Similar to cancer in IPF, pulmonary infarction is mostly characterized by a peripheral and juxta pleural well-defined opacification without air bronchograms, mostly developing in the lower lobes. The differential diagnosis is made even more difficult by the clinical presentation, characterized by dyspnoea, tachypnea, and hypoxemia that can mimic an acute exacerbation of IPF [37,38].

### 2.3. Acute Exacerbation

IPF is a progressive and chronic disease with a course that can be unpredictable. Indeed, some patients may manifest a slow and progressive deterioration of lung function, while others may present a rapid progression with a fast decline of the clinical conditions. In some cases, episodes of sudden worsening of symptoms and lung function can occur. These cases are defined as acute exacerbation (AE-IPF).

The revised definition of acute exacerbation of IPF is an acute, clinically significant, respiratory deterioration characterized by evidence of new, widespread alveolar abnormality [39]. It seriously worsens life expectancy, with up to 50% of patients dying during the acute event, and a significant decrease in short-term survival of three to four months after the episode [40,41,42,43,44,45,46]. Moreover, if the patient needs mechanical ventilation, the mortality rate increases to 90% of cases [47].

Although the pathogenesis of acute exacerbations is still unknown, it is thought to share some similarities with acute lung injury (ALI), in which a diffuse alveolar damage (DAD) occurs secondary to an acute event, such as viral infections, gastro oesophageal reflux (GER), surgical lung biopsy or resection, immunosuppressive therapy, bronchoalveolar lavage, or any mechanical procedure (Figure 12).

It has also been observed that episodes are more frequent in patients with a physiologically and functionally advanced disease, in combination with a more progressive disease [48], in wintertime [46,49,50], in young people, in patients with coronary artery disease, in cardiogenic pulmonary oedema, and in those with a high body mass index. 

The criteria to define AE-IPF are a previous or concurrent diagnosis of IPF, an unexplained worsening or development of dyspnoea within 30 days, and exclusion of alternative causes (left heart failure) [51].

On HRCT, it manifests with new bilateral ground glass abnormality and/or consolidation superimposed on a background reticular or honeycombing (Figure 13) [52]. It has been noticed that the pattern and extent of opacification can be related to the clinical outcome, observing that patients with a diffuse pattern have a higher risk of death compared to those with a peripheral one. If an asymmetrical distribution of the opacification (consolidation or ground glass) is seen on HRCT, this is a predictor of a better outcome in patients with AE-IPF [53].

HRCT is sensitive, but not specific, in recognizing AE-IPF, since other conditions may present a diffuse increase in attenuation of the lung parenchyma in a background of pre-existing fibrosis, and thus these must be taken into account in differential diagnosis. These conditions are pulmonary infections, drug-induced pneumonias, and pulmonary oedema due to heart failure.

In particular, if pleural effusion or profuse septal thickening (an uncommon feature in non-complicated IPF) are associated with the ground glass opacities, heart failure should be highly suspected [2]. Along with pathological conditions, even some technical factors during exam acquisition can determine spurious increased opacification, such as scan acquisition at the end of expiration or intravenous contrast administration [2].

A list of the main parenchymal complications occurring during IPF with their CT scan characteristics is presented in Table 1.

If a clinical suspect of AE-IPF is present, to reach the final diagnosis a multidisciplinary discussion is required.

## 3. Conclusions

HRCT has an increasingly predominant role in the diagnosis of IPF. Moreover HRCT plays a fundamental role in detecting IPF in both acute and chronic parenchymal complications that can seriously worsen the clinical course.

Besides lung cancer, patients with IPF can develop opacities, including inflammatory consolidations, areas of dense fibrosis, infarction due to pulmonary embolism, or acute exacerbation. Clinical presentation plays a crucial role for a correct interpretation of CT findings.

It is important to identify these opacities, particularly through involving an expert thoracic radiologist who can better evaluate the radiological aspects and make a comparison with previous radiological examinations when available.

## Figures and Tables

**Figure 1 medicina-55-00613-f001:**
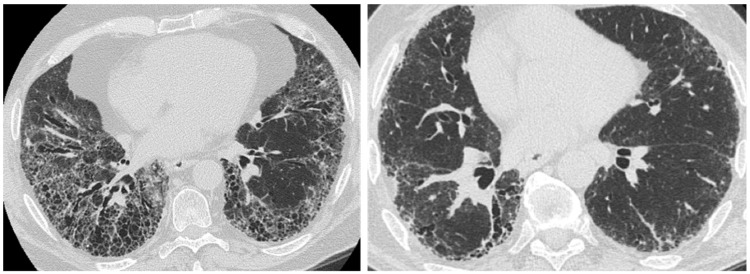
Typical usual interstitial pneumonia (UIP) pattern on High Resolution Computed Tomography (HRCT). Irregular inter- and intralobular septal thickening, with a subpleural, basal predominant. Honeycombing, traction bronchiectasis, and bronchiolectasis are clearly seen in the basal region.

**Figure 2 medicina-55-00613-f002:**
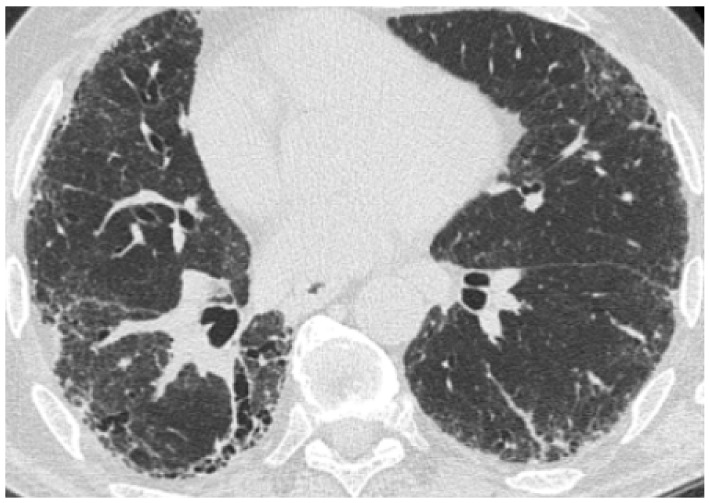
Probable UIP pattern on HRCT. Irregular inter- and intralobular septal thickening, with a subpleural, basal predominant. Traction bronchiectasis and bronchiolectasis are seen in the fibrotic areas. Honeycombing is not present.

**Figure 3 medicina-55-00613-f003:**
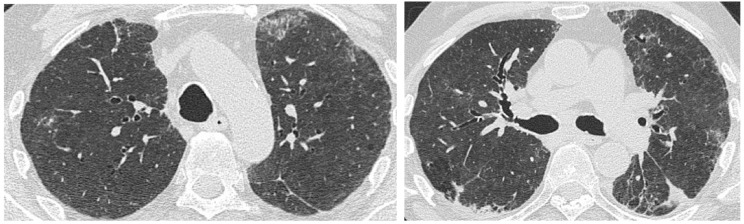
An example of *Pneumocystis jirovecii* pneumonia Broncho-Alveolar Lavage-confirmed. HRCT demonstrated patchy areas of ground glass in the upper left lobe superimposed on a fibrotic background.

**Figure 4 medicina-55-00613-f004:**
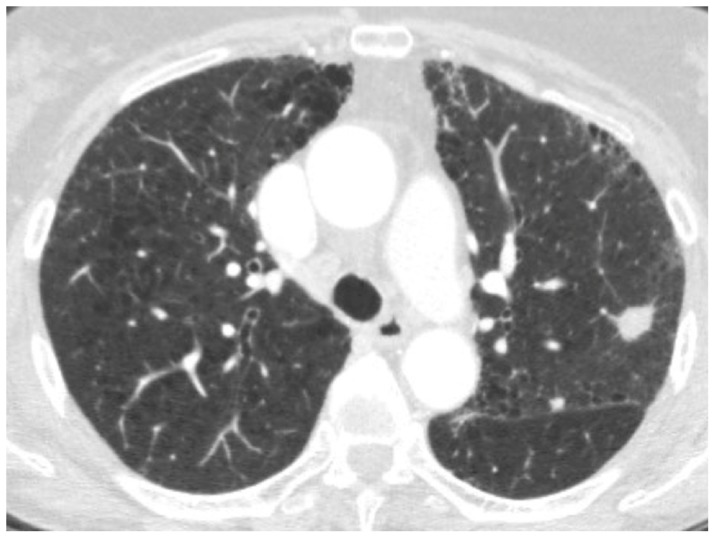
High Resolution Computed Tomography showing a nodule with spiculated margins, located in the superior left lobe. The final diagnosis after pathological examination on the surgical specimen demonstrated that it was a tuberculoma.

**Figure 5 medicina-55-00613-f005:**
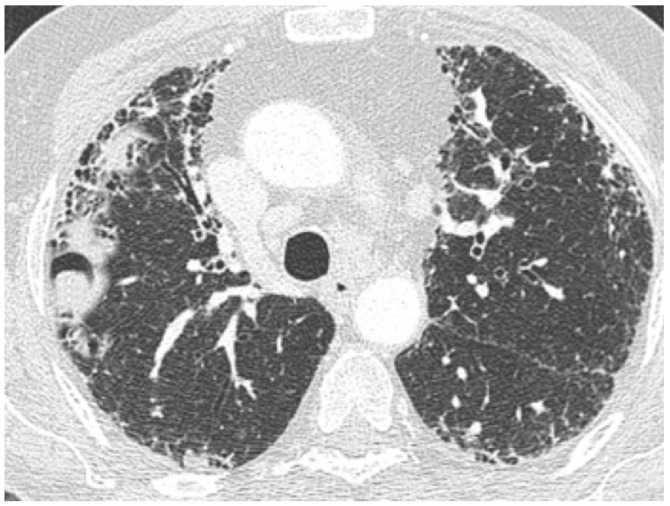
Figure courtesy of Carlo Florio. In a background of idiopathic pulmonary fibrosis, HRCT showed the typical intracavitary mass with an adjacent “air crescent sign” due to *Aspergillus* infection.

**Figure 6 medicina-55-00613-f006:**
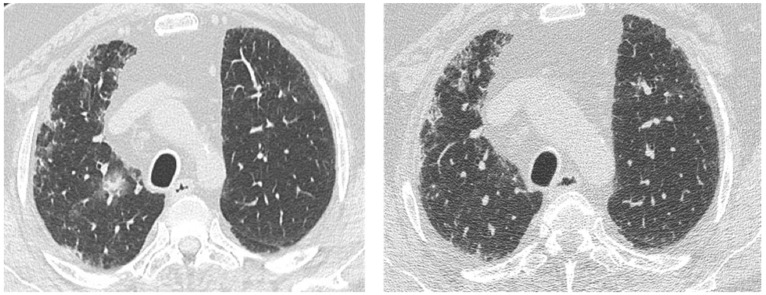
HRCT from a patient with idiopathic pulmonary fibrosis, a current smoker. (**a**) A part-solid nodule located in the upper right lobe, which, according to Fleischner Society, is highly suggestive of lung cancer; (**b**) 3 month control after corticosteroid and antibiotic therapy showing complete resolution of this opacity.

**Figure 7 medicina-55-00613-f007:**
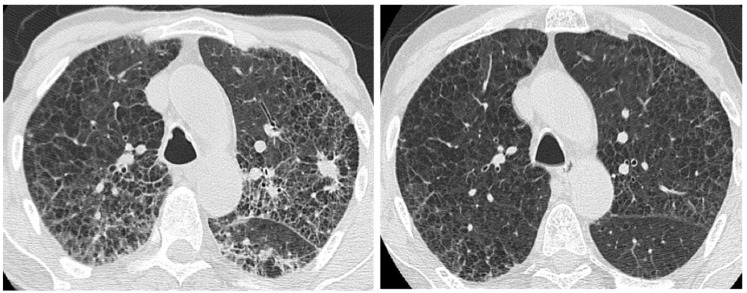
(**a**) A case of a patient with idiopathic pulmonary fibrosis with spiculated nodule; (**b**) the control after corticosteroid and antibiotic therapy showing complete resolution of this opacity.

**Figure 8 medicina-55-00613-f008:**
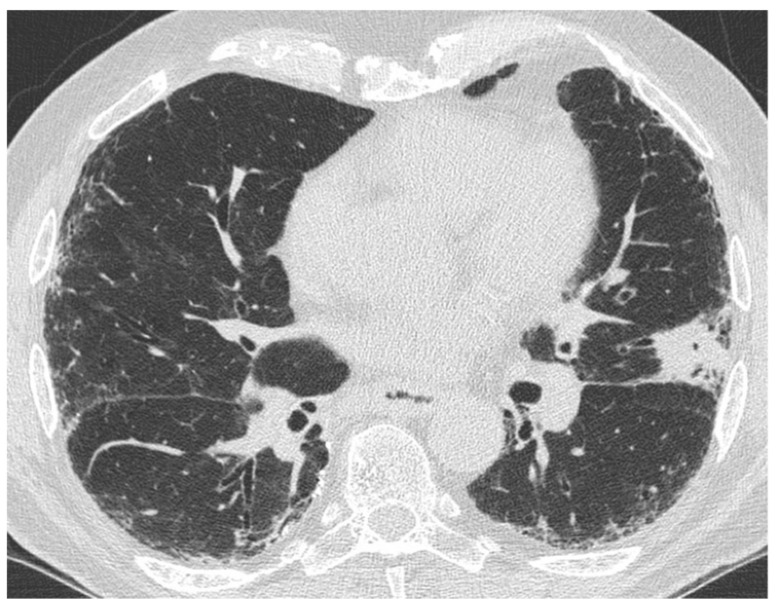
HRCT of a 70 year old man showing a consolidation area in the left upper lobe. Pathological examination after biopsy detected an adenocarcinoma.

**Figure 9 medicina-55-00613-f009:**
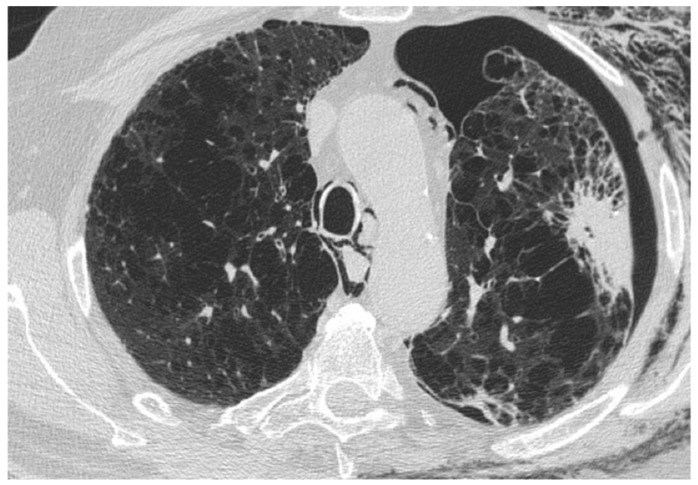
HRCT of a 63 year old man coming to the emergency room for an acute chest pain. HRCT showed, in a background of pulmonary idiopathic fibrosis, pneumomediastinum, and pneumothorax, a spiculated mass in the peripheral region of the left upper lobe due to squamous cell carcinoma.

**Figure 10 medicina-55-00613-f010:**
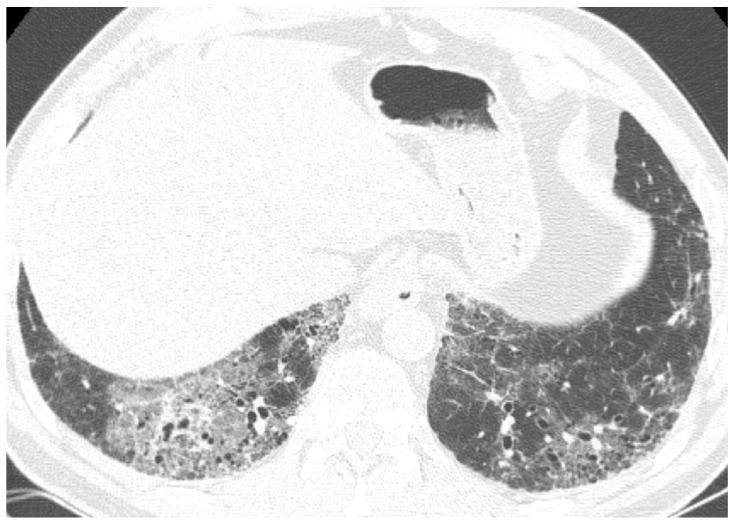
Figure courtesy of Roberta Polverosi. A well-defined ground glass opacity within a fibrosis area in the right lower lobe, persistent after corticosteroid and antibiotic therapy. The biopsy demonstrated a mucinous bronchioloalveolar carcinoma.

**Figure 11 medicina-55-00613-f011:**
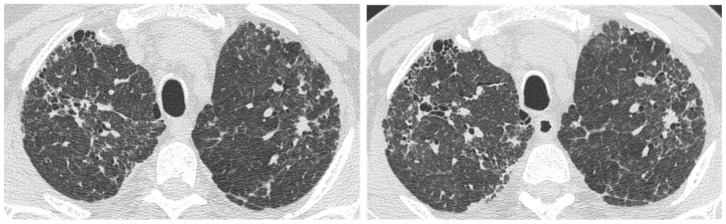
(**a**) HRCT showing a small ill-defined opacity in the upper left lobe of uncertain etiology; (**b**) the nodule remaining stable at the control HRCT after 3 years.

**Figure 12 medicina-55-00613-f012:**
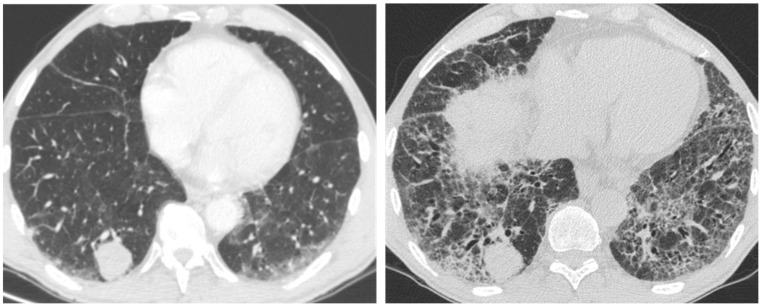
HRCT showing a rapid worsening of reticular abnormalities after a Computed Tomography-guided biopsy of a basal rounded mass.

**Figure 13 medicina-55-00613-f013:**
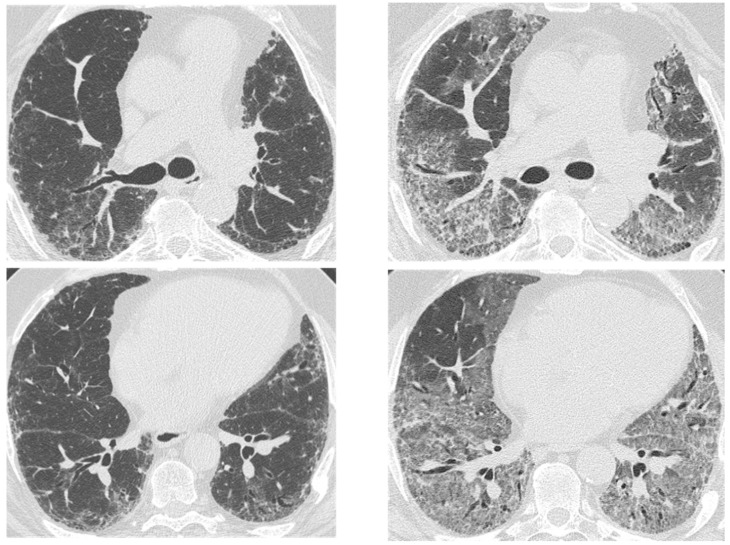
HRCT of an IPF patient with acute deterioration of symptoms and lung function, compared to the baseline HRCT. (**a**,**b**) A rapid and progressive development of diffuse and symmetric ground glass opacities; (**c**,**d**) suggestive of an acute exacerbation. The patient died 3 weeks after.

**Table 1 medicina-55-00613-t001:** Classification of lung parenchymal complications in patients with idiopathic pulmonary fibrosis (IPF) and HRCT features.

Acute/Chronic	Complication		HRCT Features
Acute	Infections	Mycobacterium species	Peripheral mass-like lesion/subpleural nodules/segmental or lobar coalescent consolidation with or without necrotising cavitation.
		*Pneumocystis jirovecii*	Ground glass opacity/no change from the baseline in a clinic context of infection.
		*Aspergillus*	Fungal fronds in a pre-existing cavity in early stages. Subsequent coalescence of the cavity (air crescent sign).
	Acute exacerbation IPF		New bilateral ground glass opacities and/or consolidation on a background of reticular or honeycombing pattern.
	Right heart failure		Profuse septal thickening, ground glass opacities, pleural effusion on a background of reticular or honeycombing pattern.
Chronic	Lung cancer		Ill-defined rounded lesion, mimicking air space consolidation/nodular lesion developing within peripheral and basal honeycombing areas. Ground glass opacity in fibrosis area (mucinous bronchioloalveolar carcinoma).

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
