# Peer review of "Imaging Review of the Lung Parenchymal Complications in Patients with IPF"

_medicina, 2019, doi:10.3390/medicina55100613_

Round 1
Reviewer 1 Report
The manuscript provides a thoughtful overview on the lung parenchymal complications in IPF. It mainly focuses on infection, malignancy and acute exacerbation.
Cooments/suggestions for improvement:
There are other causes of complications in IPF e.g. drug induced pulmonary toxicity, congestive heart failure, pulmonary embolism, pulmonary hypertension, etc. The authors could mention it in general and stated that the most common conditions will be discussed.
Line 25 needs reference.
Line 45 needs reference.
Line 52 The paper will be better if the authors mention more about bacterial, viral infection as it becomes more interest in IPF pathogenesis as well.
Line 57 needs more references and explains why Mycobacterium and Aspergillus species are common. Did the authors search enough references about infection in IPF?
Example study: Yamazaki R, et al. Clinical features and outcomes of IPF patients hospitalized for pulmonary infection:: A Japaneses cohort study. PLos One. 2016.
Line 70-75 mentioned about TB, the author needs to specify the prevalence of mycobacterial infection in IPF, reactivation in IPF patients and codes references.
Line 98 Lung cancer section, need more updated references e.g. Lung cancer in idiopathic pulmonary fibrosis: A systematic review and meta-analysis. PLos One. 2018, or Characteristics of lung cancer among patients with idiopathic pulmonary fibrosis and interstitial lung disease- analysis of instutional and population data. Respir Res. 2018.
Line 110 needs reference on survival time you mentioned and recheck if there are other larger or updated cohorts.
Line 125 GGO has been reported or less typical, needs reference.
Line 138 needs reference about the prevalence of thromboembolism in IPF.
Line 153 needs more updated references when mentioned survival time. It should not be from the review articles. The author can search to the original article from the review papers.
Line 166 reference for winter time and AE-IPF.
Author Response
We thanks the reviewer 1 for his/her constructive comments. We have followed all the suggestions for improvement
There are other causes of complications in IPF e.g. drug induced pulmonary toxicity, congestive heart failure, pulmonary embolism, pulmonary hypertension, etc. The authors could mention it in general and stated that the most common conditions will be discussed.
We have mentioned the other causes of complications in IPF
Line 25 needs reference.
ok, done
Line 45 needs reference.
ok, done
Line 52 The paper will be better if the authors mention more about bacterial, viral infection as it becomes more interest in IPF pathogenesis as well.
We have added a small paragraph on this topic
Line 57 needs more references and explains why Mycobacterium and Aspergillus species are common. Did the authors search enough references about infection in IPF?
Example study: Yamazaki R, et al. Clinical features and outcomes of IPF patients hospitalized for pulmonary infection:: A Japaneses cohort study. PLos One. 2016.
ok, done
Line 70-75 mentioned about TB, the author needs to specify the prevalence of mycobacterial infection in IPF, reactivation in IPF patients and codes references.
We have added a small paragraph and references on the prevalence of mycobacterial infections.
We did not find specific data on the reactivation rate in patients with IPF
Line 98 Lung cancer section, need more updated references e.g. Lung cancer in idiopathic pulmonary fibrosis: A systematic review and meta-analysis. PLos One. 2018, or Characteristics of lung cancer among patients with idiopathic pulmonary fibrosis and interstitial lung disease- analysis of instutional and population data. Respir Res. 2018.
ok, done
Line 110 needs reference on survival time you mentioned and recheck if there are other larger or updated cohorts.
ok, done
Line 125 GGO has been reported or less typical, needs reference.
We have rewritten the sentence and added references
Line 138 needs reference about the prevalence of thromboembolism in IPF.
ok, done
Line 153 needs more updated references when mentioned survival time. It should not be from the review articles. The author can search to the original article from the review papers.
ok, done
Line 166 reference for winter time and AE-IPF.
ok, done
Reviewer 2 Report
I have only some minor suggestions:
line 64: show no change compared to the previous
line 86: please re-write. Better to split the two concept in two different sentences.
line 155: please write better the sentence
line 167: the criteria to define AE-IPF … are
line 182: erase with
line 189: I suggest to insert here a sentence recalling Table 1. E.g.: “A list of the parenchymal complications occurring during IPF with their CT scan characteristics is presented in Table 1.
line 240: the Raghu’s reference lacks of the publication year (2004)
line 254: please remove underlining to Reference 21. Authors’ names
Author Response
We thanks the reviewer 2 for his/her constructive comments. We have followed all the suggestions for improvement
line 64: show no change compared to the previous
ok, done
line 86: please re-write. Better to split the two concept in two different sentences.
we rephrased the sentence
line 155: please write better the sentence
we rephrased the sentence
line 167: the criteria to define AE-IPF … are
ok, done
line 182: erase with
ok, done
line 189: I suggest to insert here a sentence recalling Table 1. E.g.: “A list of the parenchymal complications occurring during IPF with their CT scan characteristics is presented in Table 1.
we inserted the sentence
line 240: the Raghu’s reference lacks of the publication year (2004)
ok, done
line 254: please remove underlining to Reference 21. Authors’ names
ok, done
Round 2
Reviewer 1 Report
The authors appropriately address the questions.
Author Response
We kindly thank the reviewer for the precious suggestions.